# Potentiating TRPA1 by Sea Anemone Peptide Ms 9a-1 Reduces Pain and Inflammation in a Model of Osteoarthritis

**DOI:** 10.3390/md21120617

**Published:** 2023-11-28

**Authors:** Ekaterina E. Maleeva, Yulia A. Palikova, Viktor A. Palikov, Vitaly A. Kazakov, Maria A. Simonova, Yulia A. Logashina, Nadezhda V. Tarasova, Igor A. Dyachenko, Yaroslav A. Andreev

**Affiliations:** 1Shemyakin-Ovchinnikov Institute of Bioorganic Chemistry, Russian Academy of Sciences, ul. Miklukho-Maklaya 16/10, 117997 Moscow, Russiamarisimonova@gmail.com (M.A.S.); yulia.logashina@gmail.com (Y.A.L.); 2Branch of the Shemyakin-Ovchinnikov Institute of Bioorganic Chemistry, Russian Academy of Sciences, Prospekt Nauki, 6, 142290 Pushchino, Russia; yuliyapalikova@bibch.ru (Y.A.P.); vpalikov@bibch.ru (V.A.P.); vitalij.tomsk@list.ru (V.A.K.); dyachenko@bibch.ru (I.A.D.); 3Institute of Molecular Medicine, Sechenov First Moscow State Medical University, Trubetskaya Str. 8, Bld. 2, 119991 Moscow, Russia; compasstar@gmail.com

**Keywords:** arthritis, TRPA1, TRPV1, peptide, Ms9a-1, inflammation, osteoarthritis, non-steroidal anti-inflammatory drugs, sea anemone, *Metridium senile*

## Abstract

Progressive articular surface degradation during arthritis causes ongoing pain and hyperalgesia that lead to the development of functional disability. TRPA1 channel significantly contributes to the activation of sensory neurons that initiate neurogenic inflammation and mediates pain signal transduction to the central nervous system. Peptide Ms 9a-1 from the sea anemone *Metridium senile* is a positive allosteric modulator of TRPA1 and shows significant anti-inflammatory and analgesic activity in different models of pain. We used a model of monosodium iodoacetate (MIA)-induced osteoarthritis to evaluate the anti-inflammatory properties of Ms 9a-1 in comparison with APHC3 (a polypeptide modulator of TRPV1 channel) and non-steroidal anti-inflammatory drugs (NSAIDs) such as meloxicam and ibuprofen. Administration of Ms 9a-1 (0.1 mg/kg, subcutaneously) significantly reversed joint swelling, disability, thermal and mechanical hypersensitivity, and grip strength impairment. The effect of Ms 9a-1 was equal to or better than that of reference drugs. Post-treatment histological analysis revealed that long-term administration of Ms9a-1 could reduce inflammatory changes in joints and prevent the progression of cartilage and bone destruction at the same level as meloxicam. Peptide Ms 9a-1 showed significant analgesic and anti-inflammatory effects in the model of MIA-induced OA, and therefore positive allosteric modulators could be considered for the alleviation of OA symptoms.

## 1. Introduction

One of the significant health problems worldwide is chronic arthritis due to its high prevalence rate and the absence of efficient treatment [1,2]. The most common type of arthritis is osteoarthritis (OA), which affects more than 13% of the adult population. This is a chronic age-related disorder that leads to disability associated with pain during middle and old age. As the population ages, OA is becoming more common. OA is characterized by the progressive destruction of the articular cartilage and joint integrity [3]. Pain is the major problem for patients with osteoarthritis [4], and commonly used analgesics have moderate efficacy or cause significant side effects upon chronic administration [5,6,7]. Pain relief and improvement in quality of life are crucial outcomes of the therapy.

The joint is innervated by sensory neurons that could be activated or sensitized by various stimuli [8]. Osteoarthritis pain includes nociceptive pain induced by neural activation and sensitization, as well as tissue damage and neuropathic pain produced by damaged neurons [9]. Multiple Transient Receptor Potential (TRP) channels—are expressed in chondrocytes and sensory neurons and function as sensors responding to a broad range of stimuli [8]. Several TRP channels (e.g., Transient receptor potential ankyrin 1 (TRPA1) and Transient receptor potential vanilloid 1 (TRPV1)) are evidently associated with joint pain and inflammation.

The TRPA1 channel is widely expressed in sensory neurons and in non-neuronal cells, including chondrocytes, and participates in the development and maintenance of inflammation and pain [10,11,12]. In neurons, TRPA1 is often co-localized with TRPV1, which significantly contributes to nerve sensitization and thermal hyperalgesia [13,14]. The TRPA1 channel is involved in both the initiation/propagation of inflammatory responses and the detection of inflammatory mediators. Tissue damage leads to the release of inflammatory factors that cause activation or sensitization of the TRPA1 channel located at the endings of peptidergic nociceptive sensory neurons. The influx of Ca^2+^ causes the release of neuropeptides such as substance P (SP), neurokinin A (NKA), and calcitonin gene-related peptide (CGRP), initiating a number of signal-transduction pathways to promote the inflammatory process. Additionally, the depolarization of sensory neurons is perceived as pain in the CNS [15].

TRPA1 is expressed in human and murine OA chondrocytes and can be upregulated by pro-inflammatory cytokines such as IL-1β [11,16,17,18,19]. IL-1β and other pro-inflammatory cytokines also contribute to the generation and release of pro-inflammatory and pro-algesic molecules, including reactive oxygen species (ROS), nitric oxide (NO) and prostaglandins [20]. All these molecules were reported as TRPA1 activators [10]. TRPA1 is considered a major oxidant sensor [21] and various oxidative stress-related compounds can activate the channel during inflammation [10].

Knockout of the TRPA1 gene significantly reduces symptoms of OA in a monosodium iodoacetate (MIA)-induced model of osteoarthritis in mice [17,18]. Cell-based studies revealed TRPA1 involvement in IL-1β -induced apoptosis and increased production of the pro-inflammatory cytokine IL-6 in chondrocytes [17,22]. Taken together, the in vitro and knockout mice studies refer TRPA1 as a therapeutic target for OA treatment [23]. There are a limited number of TRPA1 antagonists accessible on the market, and the most common for research use is the compound HC030031. This compound was tested in the MIA-induced OA model and showed no effect on an index of joint discomfort in the MIA-treated paw [24]. However, injection of HC-030031 (100 μM 50 μL) into the OA knee joint decreased the frequency of spontaneous excitatory synaptic current (sEPSC) induced by allyl isothiocyanate (AITC) in the substantia gelatinosa of rats with knee OA. TRPA1 inhibition also attenuated pain-related behavior in rats with knee joint OA (improved the threshold of direct pressure stimulation, increased standing time and max contact area of rat paws), while no difference was observed between AITC and saline administration [25]. Another TRPA1 antagonist, ALGX-XC20 (30 mg/kg; oral), significantly reversed (~50%) the MIA-evoked reduction in weight-bearing distribution, while the anti-allodynic effect of ALGX-XC20 was less pronounced [26].

The peptide Ms 9a-1 was isolated from the sea anemone *Metridium senile* and characterized as a positive modulator of the TRPA1 channel. Ms 9a-1 significantly potentiated agonists-induced currents of TRPA1 expressed in *X. laevis* oocytes and DRG neurons [27]. Intravenous injection of Ms 9a-1 (0.1–0.3 mg/kg) reduced pain induced by AITC (selective TRPA1 agonist), effectively reversed CFA-induced thermal hyperalgesia, and decreased CFA- and AITC-induced paw edema in mice [27,28]. The mechanism of Ms 9a-1 analgesic action includes activation of TRPA1 on sensory neurons following desensitization of TRPA1-expressing peptidergic nociceptors. Pretreatment with a TRPA1 antagonist completely reverses the analgesic effect of the peptide in mice. The same effect was reported for parthenolide, the active compound of feverfew (*Tanacetum parthenium*), which is a weak activator of TRPA1 [29]. Ms9a-1 was found to be safe; injection of peptide (10 µL of ~70 µM solution) into the hind paw did not cause pain, inflammation, or hypersensitivity, and no changes in locomotor activity in the open-field test were found after intravenous administration of 0.3 mg/kg [27].

In this work, we analyzed the ability of Ms 9a-1 to suppress joint inflammation and inhibit thermal and mechanical hyperalgesia, associated with MIA-induced OA. Peptide Ms 9a-1 significantly reversed inflammation-related behavior and reduced histological changes in MIA-injected joints.

## 2. Results

### 2.1. Preliminary Experiments

For the first time, we evaluated the potential of Ms 9a-1 in a model of MIA-induced arthritis compared to peptide APHC3, a polypeptide modulator of the TRPV1 channel from the sea anemone *Heteractis crispa* [30]. APHC3 (0.01–0.1 mg/kg) significantly reduced inflammation and pain-related behavior in models of osteo- and rheumatoid arthritis [31]. Ms 9a-1 (0.1 mg/kg) showed similar effects to APHC3 (0.1mg/kg). The peptide Ms9a-1 significantly improved the ability of animals to use arthritic limbs, reverting grip strength, articular discomfort (Figure 1) and mechanical allodynia (Figure A1) to norm after a single dosage (day 3) and regular administration (day 7).

### 2.2. MIA-Induced Arthritis

Based on preliminary experiments, we conducted a complex investigation of Ms 9a-1 in MIA induced arthritis model at a 0.1 mg/kg dose. Assessments of inflammation in vivo and functional tests for thermal and mechanical hypersensitivity and pain-induced articular disability were conducted on days 3, 7, and 14 after OA induction with MIA injection into the right knee joint. Histological changes were evaluated on days 8, 15 and 29 (Figure 2).

#### 2.2.1. Assessment of Inflammation In Vivo

Articular inflammation in vivo was assessed by the increase in injected joint diameter. The ratio of injected to intact joint diameters (in percent of the intact joint) was calculated. The most prominent increase in the ratio of injected to intact joint diameter was observed in the group administered with saline after MIA injection. In this group, the ratio was significantly greater than in the control group on days 3–14. The increase was 20% on day 3, with a subsequent decline to 10% on day 14. Similar dynamics were observed in groups administered with Ms 9a-1 and both reference drugs. Administration of 0.1 mg/kg Ms 9a-1 markedly reduced joint inflammation even after the first injection on day 3 compared to the saline-treated group. Meloxicam effectively reduced joint inflammation on days 7 and 14 as compared to the saline-treated group, while in the ibuprofen-treated animals it was significantly lower only on day 3 (Figure 3).

#### 2.2.2. Assessment of Pain-Related Behavior

Sensitivity changes and articular dysfunction related to pain sensation were assessed. Changes in mechanical and thermal sensitivity after OA induction were tested with von Frey filaments and a hot plate, respectively. Von Frey method detects pain threshold as the paw withdrawal response. MIA injection resulted in mechanical allodynia manifested as an apparent (30–40%) decrease in hind paw withdrawal threshold in the saline-treated group (Figure A2). We used the difference (Δ) between the nociceptive thresholds of ipsilateral and contralateral paws to compare effects of treatment. A decrease innociceptive threshold Δ corresponds to a higher sensitivity of the paw with OA. Ms 9a-1 administration led to a significant increase in paw withdrawal threshold as compared to saline-treated animals. Animals treated with Ms 9a-1 did not differ from the control group throughout the observation period. Both reference drugs also significantly increased the nociceptive threshold on days 3 and 7 compared to the saline-treated group (Figure 4).

The hot plate test revealed thermal hypersensitivity, indicated by a latency decrease, in the saline-treated group on days 3 and 7 as compared to the control. On the 14th day, the saline-treated animals did not significantly differ from the control. Ms 9a-1 and both reference drugs alleviated thermal hyperalgesia, as compared to the saline-treated group. It is worth noting that Ms 9a-1 was significantly effective throughout the observation period, meloxicam on days 3 and 7, while ibuprofen only on day 7 (Figure 5). 

#### 2.2.3. Assessment of Pain-Induced Articular Dysfunction

We analyzed pain-induced articular dysfunction in the incapacitance test and grip strength test. Incapacitance tester helps evaluate articular discomfort related to pain sensation. It measures the weight-bearing differences between arthritic and intact hind limbs. In control animals, weight is distributed equally between two paws. In all groups of MIA-injected animals, weight redistribution was observed with the unloading of the arthritic limb. The most prominent differences between intact and injected limbs, about 40%, were observed in saline and meloxicam treated groups on days 3 and 7. Ms 9a-1 treatment significantly improved articular functionality. Arthritic limb loading in Ms 9a-1-treated animals was restored and did not significantly differ from the control group throughout the observation period. The first administration of Ms 9a-1 on day 3 effectively reversed pain-induced knee joint incapacitation as compared to the saline-treated group. It is worth noting that neither meloxicam nor ibuprofen showed similar results on day 3. Ibuprofen slightly relieved inflammation-induced discomfort only on day 14, and meloxicam was unable to significantly improve articular functionality throughout the assessed interval (Figure 6).

The functional disability of the arthritic hind limb was estimated in grip strength test. Arthritis induction followed by saline treatment led to a significant grip strength deficit on days 3–14, constituting approximately 50–60% of the control group level. At the same time, Ms 9a-1 treatment significantly improved arthritic limb functionality from the first administration on day 3 to the last administration on day 14, as compared to the saline-treated group. Ibuprofen was also effective in preventing grip strength deficiency from the first administration on day 3, while meloxicam was not. On days 7 and 14, both reference drugs were effective, and grip strength in these groups did not differ from the control group (Figure 7).

#### 2.2.4. Knee Joint Histology

We collected samples of injected knee joints on days 8, 15 and 29 after arthritis induction for the analysis of characteristic histological signs of OA. We estimated synovial inflammation and hyperplasia, destruction of the articular cartilage and bone tissues. No pathomorphological changes in the intra-articular structures were found in a group of control animals on the 8th and 29th days after a single injection of saline into the joint cavity. A single case with minimally expressed signs of synovitis was found on the 15th day of observation as a result of trauma to the synovium during intra-articular injection of saline solution. On the 8th day after OA induction with a single intra-articular injection of MIA, pronounced inflammatory infiltration of the synovial membrane, synovial hyperplasia, moderate destructive changes in articular cartilage, and initial destructive changes in bone tissue were found in all animals. The administration of non-steroidal anti-inflammatory drugs (meloxicam and ibuprofen) or Ms 9a-1 peptide reduced pathomorphological changes in joints (Figure 8 and Figure 9).

On the 15th day of the experiment in saline-treated animals, synovitis was approximately at the same level as on the 8th day of observation, but there was a significant increase in destructive changes in cartilage and bone tissue (Figure 9). The administration of meloxicam and ibuprofen reduced the inflammatory infiltration of the synovial membrane, which positively affected the degree of synovial hyperplasia. Destructive changes in cartilage tissue of animals treated with reference drugs, however, did not differ significantly from animals treated with saline. Nevertheless, destructive changes in bone tissue were less pronounced in rats after meloxicam treatment. All indicators were more favorable in animals treated with Ms 9a-1 than in saline-treated animals.

On the 29th day of observation, the signs of arthritis continued to progress in saline-treated animals: synovitis and synovial hyperplasia were evident, while changes in cartilage and bone tissue came to the fore. Therapy with meloxicam from days 3 to 14 helped to reduce the severity of synovitis and synovial hyperplasia, but it was less effective against the destruction of cartilage and bone tissue. The administration of Ms 9a-1 from days 3 to 14 resulted in a similar effect to that of meloxicam, namely a reduction in the severity of synovitis and synovial hyperplasia, as well as a reduction in the destructive processes in cartilage and bone tissue. According to the histological study, peptide Ms 9a-1 was more effective than ibuprofen and had comparable pharmacological effects to meloxicam in the MIA-induced OA model (Figure 8 and Figure 9).

Representative images illustrating histological analysis of joints for each group on day 15 day 29 are shown in Figure 10 and Figure A3, respectively.

## 3. Discussion

Osteoarthritis (OA) is the most common form of arthritis and a major source of pain and disability in the adult population [32,33,34]. Inflammation-induced hyperalgesic responses could be reduced by analgesics that decrease nociceptors sensitization and spontaneous activation [35]. TRPA1 is widely expressed in neurons and other cells; it acts as a sensor for oxidative stress-related compounds, and inflammatory stimuli could increase the expression of TRPA1 [10,11,36]. TRPA1 mediates inflammatory pain and plays a significant role in the development of arthritis [37], including the model of MIA-induced arthritis, as shown in knockout mice and TRPA1 antagonists’ treatment [17,18]. It is intriguing that TRPA1 could be an effective therapeutic target for inflammatory pain in older populations [38]. It is generally admitted that TRPA1 channel is one of the attractive targets for alleviation of OA symptoms, and at least one TRPA1 antagonist is undergoing clinical trials for OA treatment [11,19,22,39,40,41].

In the present work, we analyzed the efficacy of peptide Ms 9a-1 in the MIA-induced model of OA as an anti-inflammatory and analgesic compound. Ms 9a-1 is the positive modulator of the TRPA1 channel that produces significant anti-inflammatory and analgesic effects in vivo [27,28]. Positive modulators and weak activators of TRPA1 could produce desensitization of the channel or defunctionalization of TRPA1-expressing sensory neurons to other stimuli that result in significant anti-inflammatory and analgesic effects [29,42,43,44,45]. TRPA1 usually co-expresses with TRPV1 on sensory neurons and the effect of positive modulators and weak activators of TRPA1 partially overlaps with inhibition or desensitization of TRPV1, including attenuation of thermal hyperalgesia [46]. Therefore, the pharmacological effects of such compounds are distinct from those of TRPA1 antagonists. Nothing is known about the effects of positive modulators and weak activators on other cells expressing TRPA1, such as chondrocytes. Furthermore, none of the positive modulators of TRPA1 have been tested in the model of MIA-induced arthritis before.

The injection of MIA into the knee joints causes disturbance of cellular glycolysis and chondrocyte death, leading to cartilage and bone destruction that mimics certain aspects of OA such as joint pathology progression and pain-related behavior [41,47,48]. This model produces a rapid, reproducible, robust pain-like phenotype and extensive joint pathology, which does not correspond to the slow development of human OA [41,47,48]. Nevertheless, this model is considered suitable for testing analgesic and anti-inflammatory compounds but provides no information on the influence of these compounds on OA pathogenesis [41]. MIA-induced OA is characterized by pain-induced functional disability accompanied by hypersensitivity to thermal and mechanical stimuli [31,47]. We found that Ms 9a-1 significantly alleviated MIA-induced arthritic symptoms such as joint swelling, pain-induced behavior, and hypersensitivity to the various stimuli in rats.

We compared Ms 9a-1 (positive modulator of TRPA1) with APHC3 (a modulator of the TRPV1 channel), and non-steroidal anti-inflammatory drugs (NSAIDs) such as meloxicam (a selective cyclooxygenase (COX)-2 inhibitor) and ibuprofen (a non-selective COX inhibitor). NSAIDs are often insufficient to relieve pain, but they are still the most commonly recommended and used drugs in OA treatment [6,7]. Different COX inhibitors can produce variable effects on induced arthritis in rats [31,49] that could be the result of the pharmacological action of their metabolites [49]. Peptide APHC3 inhibits the TRPV1 channel on sensory neurons and consequently reduces neurogenic inflammation [30,31,50]. Thus, both APHC3 and Ms 9a-1 produce an effect on the excitability of sensory neurons via different molecular targets.

In our study, we started treatment on day 3 after the MIA injection and assessed joint inflammation and pain-related behavior 60 min after the first-time administration of compound or saline, which reflects the effect of a single dose. The first injection of Ms 9a-1 at 0.1 mg/kg significantly reversed joint swelling, supporting the important role of neurogenic inflammation maintained by TRPA1-expressing neurons in this process. The doses of meloxicam and ibuprofen corresponded to the maximum recommended doses for human treatment [6,7,51]. Ibuprofen also showed a significant effect of single dose, but meloxicam was unable to reduce the inflamed joint diameter after the first administration. (Figure 3). A single dose of Ms 9a-1 produced a significant analgesic effect to reverse hypersensitivity, functional disability and weakening of grip strength associated with movement-induced pain. Meloxicam and ibuprofen showed variable effects in different tests and timepoints (Figure 4, Figure 5, Figure 6 and Figure 7), which could be a reason for the unsatisfactory feedback of patients after the usage of these drugs in OA treatment [6,7,51].

Further, we analyzed the effects of regular administration of compounds on day 7 (5 days of the treatment) and day 14 (12 days of the treatment). Ms 9a-1 and meloxicam significantly reduced joint diameter on days 7 and 14, confirming their anti-inflammatory properties in progressive joint destruction, when ibuprofen did not show a significant effect (Figure 3). Regular administration of all compounds did not improve functional disability compared to the saline-treated group, but Ms 9a-1 decreased shift in weight-bearing, making it non-significant compared to the control group. Ms 9a-1 and meloxicam significantly reversed all behavioral hallmarks (mechanical allodynia, grip strength impairment, and thermal sensitivity) of joint inflammation on days 7 and 14, while ibuprofen showed non-stable effects.

Changes in articular cartilage structure and function during OA lead to joint pain and structural changes, affecting mobility [37]. Therefore, we analyzed histological changes in MIA-induced arthritis after treatment with COX inhibitors and the peptide Ms 9a-1. We chose two timepoints in the acute phase of joint inflammation and destruction (day 8 and day 15) and an additional timepoint—day 29—two weeks after treatment, to analyze the further progression of pathological changes. All compounds showed no statistically significant effect compared to the saline-treated group. Nevertheless, tested compounds reduced synovitis, synovial hyperplasia and bone destruction to non-significant levels compared to the control group on day 8. Meloxicam additionally reduced synovitis and bone destruction on day 15. Interestingly, in the post-treatment period (day 29), administration of Ms 9a-1 or meloxicam prevented further progression of articular cartilage and bone destruction, while ibuprofen had no positive effect.

Ongoing pain after joint damage is dependent on afferent fiber activity that is only partially initiated by TRPV1 or TRPA1 activation [24,52]. Blockade of the TRPA1 channel reduced the responses of WDR neurons to high intensity stimulation, but did not decrease spontaneous firing in the MIA-induced pain model [52]. Therefore, inhibition of only TRPV1 or TRPA1 could be not enough for efficient alleviation of arthritis pain [24]. Nevertheless, knockout of the TRPA1 gene and inhibitors of TRPA1 reduce the development of inflammation and pain response in MIA-induced arthritis [18,22,40,53]. Potentiating TRPA1 by the peptide Ms 9a-1 produced significant analgesic and anti-inflammatory effects in MIA-induced arthritis. Additionally, Ms 9a-1 effectively reversed thermal hyperalgesia and reduced histological changes in the damaged joints. Similar effects on MIA-induced arthritis were observed after administration of APHC3 (Figure 1) [31]. APHC3 is the peptide modulator of the TRPV1 channel and can either potentiate or inhibit the response of the channel depending on the strength of the activation stimuli [50], and it has significant analgesic and anti-inflammatory activity in different in vivo and ex vivo models [30,54]. Therefore, we can suggest that Ms 9a-1 affects TRPV1/TRPA1-expressing sensory neurons and decreases neurogenic inflammation, which leads to a positive outcome. We did not find any additional benefits or disadvantages of TRPA1 modulation on chondrocytes compared to reference NSAIDs or modulation of TRPV1 by the peptide APHC3.

## 4. Materials and Methods

### 4.1. Ethics Statement

All experiments in this study conform fully to the World Health Organization’s International Guiding Principles for Biomedical Research Involving Animals. Experimental protocol was approved by the Institutional Commission for the Control and Use of Laboratory Animals of the Branch of the Shemyakin-Ovchinnikov Institute of Bioorganic Chemistry of the Russian Academy of Sciences (protocol number: 805/21, date of approval: 18 June 2021).

### 4.2. Drugs

Ms 9a-1 was produced as described previously [27]. Meloxicam and ibuprofen were purchased from Sigma-Aldrich (Moscow, Russia).

### 4.3. Animals

Sprague Dawley 8–10-week-old male rats weighing 250–270 g were used in this study. Animals were obtained from Animal Breeding Facility Branch of Shemyakin-Ovchinnikov Institute of Bioorganic Chemistry, Russian Academy of Sciences, Pushchino, Russia). Rats were housed under controlled conditions at room temperature (23 ± 2 °C) in a 12 h light–dark cycle and provided ad libitum access to food and water.

### 4.4. Model of MIA-Induced Osteoarthritis and Compound Administration

To induce osteoarthritis with MIA, on day 0 rats were anesthetized with an intramuscular injection of Zoletil (20–40 mg/kg, Virbac Sante Animale) and Xylazine (5–10 mg/kg, Pharmamagist, Ltd., Budapest, Hungary). Intra-articular injection of 3 mg MIA in 50 µL of sterile saline to the right knee joint was carried out to all groups of animals except for the control group. The control animals were injected with 50 µL of sterile saline. The left joint remained intact in all groups.

Rats of all groups were treated with test compounds from day 3 to day 14 after MIA injection (Figure 2). Saline or Ms 9a-1 (0.1 mg/kg) were injected subcutaneously (2 mL/kg), meloxicam (1 mg/kg) and ibuprofen (40 mg/kg) were gavaged (2 mL/kg).

Pain-related behavior was tested 60 min after administration of test compounds on days 3, 7, and 14.

### 4.5. Assessment of Inflammation In Vivo

To evaluate the degree of swelling after MIA injection, we measured the knee joint diameters of both legs using a digital caliper. The joint diameter ratio was calculated as (diameter of injected joint/diameter of intact joint) × 100.

### 4.6. Assessment of Pain-Related Behavior

#### 4.6.1. Hot Plate Test

MIA-induced arthritis is accompanied by hypersensitivity to heat [24,55]. Hot-Plate Analgesia Meter (Columbus Instruments, Columbus, OH, USA) set at 55 °C was used for pain threshold assessment. Each rat was placed on the preheated hot-plate surface until a nociceptive reaction (withdrawal or licking of the hind paw) was detected. The test was discontinued after 30 s if no withdrawal response was observed. Pain threshold was specified as the latency to hind paw licking or withdrawal.

#### 4.6.2. Hind Limb Grip Strength Test

A bilateral hind limb grip strength test was used to evaluate movement-evoked pain. Three measurements were conducted on a Grip Strength Meter (Columbus Instruments, Columbus, OH, USA) with an interval of 30 s for the calculation of the mean grip strength. During the test, a gently restrained rat was allowed to grab the wire mesh frame of the apparatus with hind paws. Then the rat was pulled backwards, and maximal force applied to the frame just before grip release was recorded.

#### 4.6.3. Test with Von Frey Filaments

Electronic von Frey filament (BIO-EVF, Bioseb, Vitrolles, France) was used to analyze cutaneous mechanical sensitivity. Rats had an adaptation period (10 min) in plastic cages with an elevated mesh bottom, allowing access to the plantar surface of the hind paws. A filament was applied perpendicularly to the plantar surface of the hind paw, and the pressure evoked paw withdrawal reflex was recorded. Three trials with inter-application intervals of 1 min to avoid sensitization to the mechanical stimuli were conducted. Δ nociceptive threshold was calculated as (nociceptive threshold of the ipsilateral paw - nociceptive threshold of the contralateral paw).

#### 4.6.4. Incapacitation Test

The weight distribution between rear paws reflects the level of discomfort in the arthritic limb. Rats were allowed to adapt for at least 5 min in the chamber of the incapacitance tester (BIO-SWB-TOUCH, Bioseb, Vitrolles, France) before the weight on each hindlimb was measured. We conducted three trials with intervals of at least 1 min for calculation of the mean weight distribution value. Data for the arthritic limb were normalized to the intact limb. Equal weight distribution corresponds to ~100%, and it was <100% in the MIA-induced model of arthritis.

### 4.7. Joint Histology

Biomaterial for histological examination was collected after the euthanasia of experimental animals. Soft tissues around the right knee joint were excised as much as possible; the femur, tibia and fibula were cut transversely in the middle of the diaphysis, closer to the corresponding articular surfaces. The joints were fixed in a 10% solution of neutral formaldehyde for 7 days, washed by water and subjected to subsequent decalcification in Trilon B for 10–14 days. After decalcification of bone and cartilaginous tissue, the joints were cut in the sagittal plane, and the diaphyses of the tubular bones were shortened to a border of 2–3 mm from the metaepiphyseal cartilage. Then, the biomaterial was washed by water, dehydrated in alcohols of increasing concentration, and embedded in paraffin. Paraffin sections 4–5 µm thick were stained with hematoxylin and eosin, as well as hematoxylin—strong (fast) green—safranin O, and studied using conventional light microscopy on an AxioScope.A1 microscope (Carl Zeiss, Oberkochen, Germany). Microphotographs of histological preparations were obtained using a high-resolution camera Axiocam 305 color (Carl Zeiss, Oberkochen, Germany) and ZEN 2.6 lite software (Carl Zeiss, Oberkochen, Germany). Histological analysis assessed the following morphological signs: inflammatory infiltration of the synovial membrane (synovitis), synovial hyperplasia, destructive changes in articular cartilage, and destructive changes in bone tissue. The 5-point scale was used to assess the degree of severity of a particular morphological feature, where 0 represents normal tissue and 5 represents severe tissue degeneration [56].

### 4.8. Statistical Analysis

Statistical analysis was performed in GraphPad Prism 6.0 for Windows (GraphPad Software, San Diego, CA, USA). The non-parametric Kruskal–Wallis test with Dunn’s multiple comparisons post-test was applied for multiple independent samples. A statistically significant difference was considered at a value of *p* < 0.05. On the box plots, data are presented as median, mean (+), first to third interquartile range, minimum, and maximum. Histograms show the mean with a standard deviation (SD).

## 5. Conclusions

Subcutaneous injection of the peptide Ms 9a-1 at a dose of 0.1 mg/kg significantly reversed inflammation and pain-related behavior in rats with MIA-induced OA. The efficacy of Ms 9a-1 was higher or equal to that of commonly used NSAIDs (meloxicam and ibuprofen). The significant anti-inflammatory and analgesic effects were observed within 1 h after peptide administration and were accompanied by the recovery of hind limb lability and functionality. A decrease in inflammation showed a positive effect on the histological score of cartilage and bone destruction. Thus, positive modulators of TRPA1, such as the peptide Ms 9a-1, could be considered as pharmacological agents for the treatment of arthritis symptoms.

## Figures and Tables

**Figure 1 marinedrugs-21-00617-f001:**
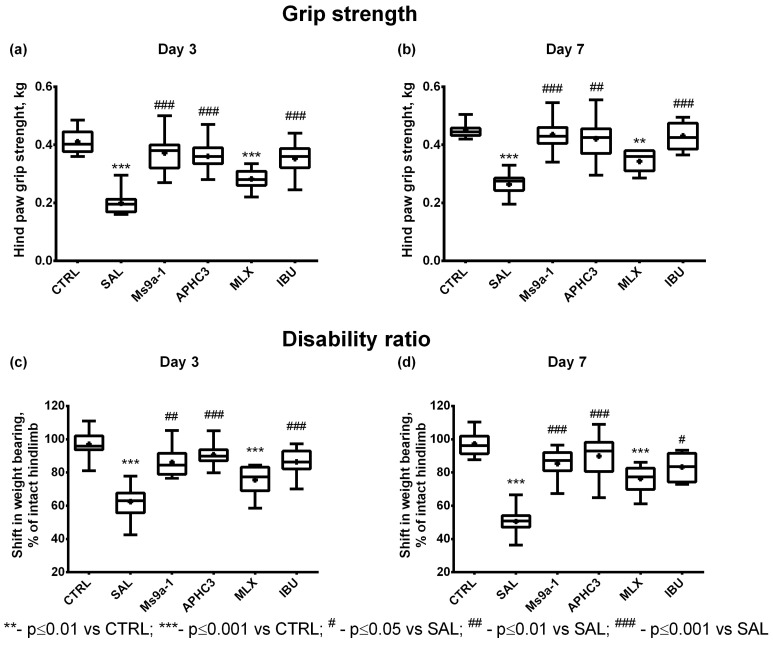
Analgesic activity of peptides APHC3 and Ms 9a-1 compared to non-steroidal anti-inflammatory drugs in the MIA-induced OA model. Grip strength was assessed with a Grip Strength Meter on days 3 (**a**), and 7 (**b**), after OA induction. Weight distribution between rear paws was estimated with the incapacitance tester on days 3 (**c**), and 7 (**d**) after intra-articular MIA injection. OA was induced by intra-articular MIA injection into the right knee joint (3 mg MIA in 50 μL of sterile saline). Control rats (CTRL) received intra-articular injection of 50 μL of sterile saline into the right knee joint. Ms9a-1 (0.1 mg/kg s.c.), APHC3 (0.1 mg/kg s.c.), meloxicam (0.5 mg/kg i.m.), and ibuprofen (40 mg/kg p.o.) were administered daily on days 3–7. CTRL—control group, SAL—sterile saline, MLX—meloxicam, and IBU—ibuprofen. Results are shown as median with mean marked as a cross (+), interquartile range, minimum, and maximum (*n* = 10 for each group except control, where *n* = 12). Statistical analysis was performed using the Kruskal–Wallis test followed by Dunn’s multiple comparisons test. ** *p* < 0.01 vs. CTRL, *** *p* < 0.001 vs. CTRL, **#**
*p* < 0.05 vs. SAL.

**Figure 2 marinedrugs-21-00617-f002:**
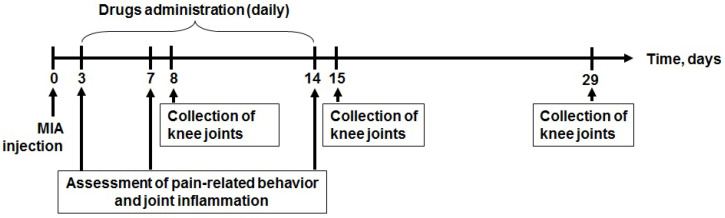
Experimental timeline for the MIA-induced arthritis model.

**Figure 3 marinedrugs-21-00617-f003:**
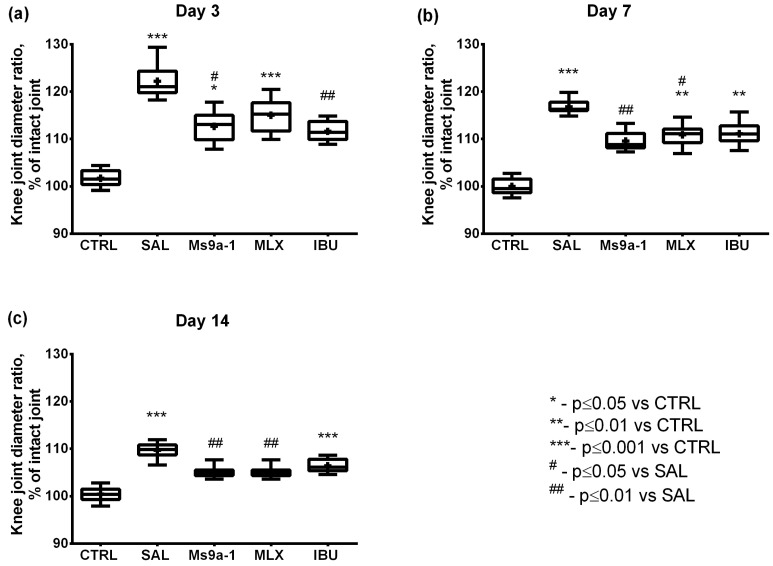
Normalized diameters of the knee joints in the MIA-induced osteoarthritis model on days 3 (**a**), 7 (**b**) and 14 (**c**). Knee joint diameter ratios are expressed in percent of intact joint diameters. Ms 9a-1 (0.1 mg/kg s.c.), meloxicam (0.5 mg/kg i.m.), and ibuprofen (40 mg/kg p.o.) were administered daily on days 3–14. CTRL—control group, SAL—sterile saline, MLX—meloxicam, and IBU—ibuprofen. Results are shown as median with mean marked as a cross (+), interquartile range, and minimum and maximum (*n* = 10 for each group). Statistical analysis was performed using the Kruskal–Wallis test followed by Dunn’s multiple comparisons test. * *p* < 0.05 vs. CTRL, ** *p* < 0.01 vs. CTRL, *** *p* < 0.001 vs. CTRL, # *p* < 0.05 vs. SAL, ## *p* < 0.01 vs. SAL.

**Figure 4 marinedrugs-21-00617-f004:**
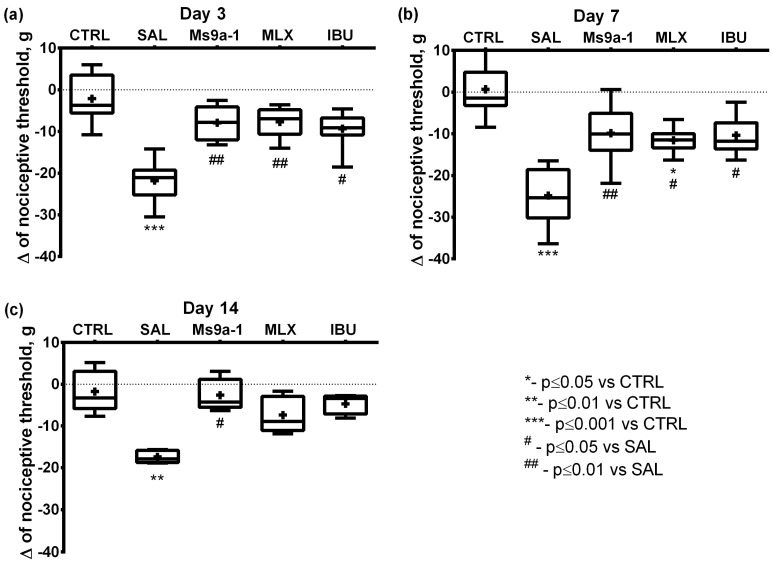
Mechanical allodynia in the MIA-induced OA model was assessed on days 3 (**a**), 7 (**b**), and 14 (**c**) after intra-articular MIA injection into the right knee joint. Change in nociceptive threshold was calculated as Δ = (nociceptive threshold of ipsilateral paw − nociceptive threshold of contralateral paw). CTRL—control group, SAL—sterile saline, MLX—meloxicam, and IBU—ibuprofen. Results are shown as median with mean marked as a cross (+), interquartile range, minimum, and maximum (*n* = 10 for each group). Statistical analysis was performed using the Kruskal–Wallis test followed by Dunn’s multiple comparisons test. * *p* < 0.05 vs. CTRL, ** *p* < 0.01 vs. CTRL, *** *p* < 0.001 vs. CTRL, **##**
*p* < 0.01 vs. SAL.

**Figure 5 marinedrugs-21-00617-f005:**
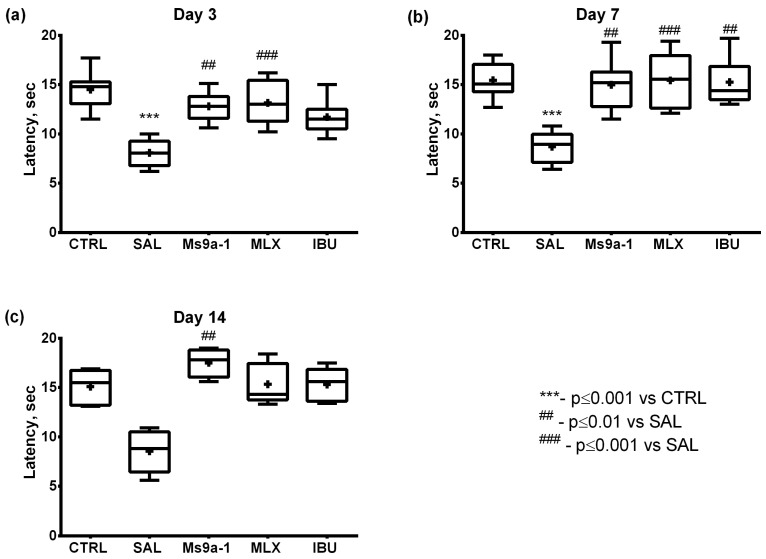
Thermal sensitivity in the MIA-induced OA model. Paw withdrawal latency was assessed in the hot plate test on days 3 (**a**), 7 (**b**), and 14 (**c**) after intra-articular MIA injection. CTRL—control group, SAL—sterile saline, MLX—meloxicam, and IBU—ibuprofen. Results are shown as median with mean marked as a cross (+), interquartile range, minimum, and maximum (*n* = 10 for each group). Statistical analysis was performed using the Kruskal–Wallis test followed by Dunn’s multiple comparisons test. *** *p* < 0.001 vs. CTRL, **##**
*p* < 0.01 vs. SAL, **###**
*p* < 0.001 vs. SAL.

**Figure 6 marinedrugs-21-00617-f006:**
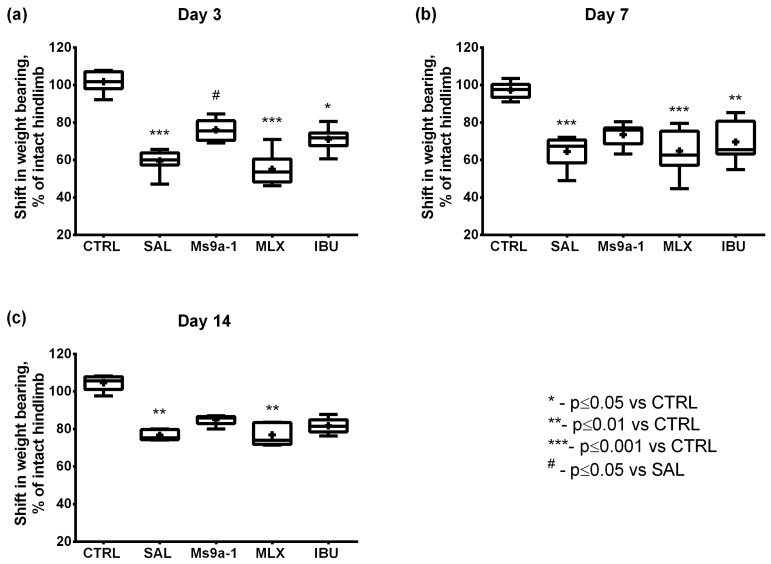
Normalized level of discomfort in the arthritic limb in the MIA-induced OA model. Weight distribution between rear paws was estimated on days 3 (**a**), 7 (**b**), and 14 (**c**) after intra-articular MIA injection. Results are shown as median with mean marked as a cross (+), interquartile range, minimum, and maximum (*n* = 10 for each group). Statistical analysis was performed using the Kruskal–Wallis test followed by Dunn’s multiple comparisons test. * *p* < 0.05 vs. CTRL, ** *p* < 0.01 vs. CTRL, *** *p* < 0.001 vs. CTRL, **#**
*p* < 0.05 vs. SAL.

**Figure 7 marinedrugs-21-00617-f007:**
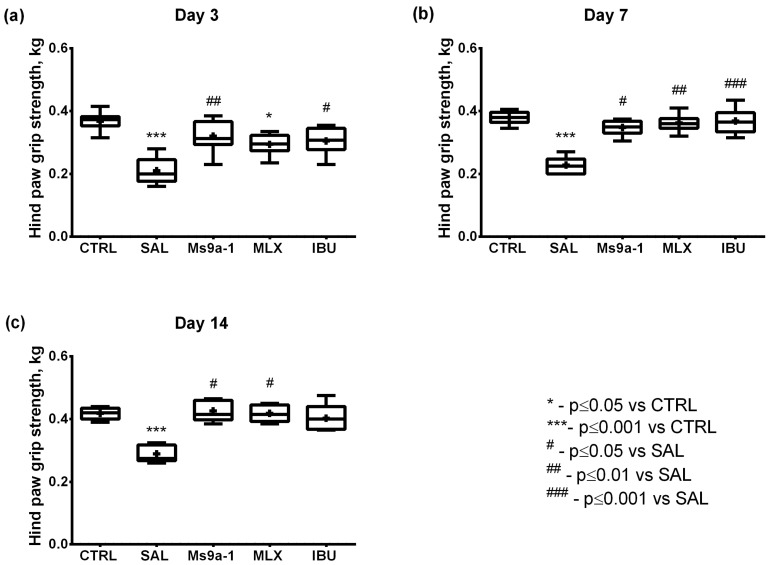
Grip strength of the arthritic limb in the MIA-induced OA model. Grip strength was assessed on days 3 (**a**), 7 (**b**), and 14 (**c**) after intra-articular MIA injection. CTRL—control group, SAL—sterile saline, MLX—meloxicam, and IBU—ibuprofen. Results are shown as median with mean marked as a cross (+), interquartile range, minimum, and maximum (*n* = 10 for each group). Statistical analysis was performed using the Kruskal–Wallis test followed by Dunn’s multiple comparisons test. * *p* < 0.05 vs. CTRL, *** *p* < 0.001 vs. CTRL, **#**
*p* < 0.05 vs. SAL, **##**
*p* < 0.01 vs. SAL, **###**
*p* < 0.001 vs. SAL.

**Figure 8 marinedrugs-21-00617-f008:**
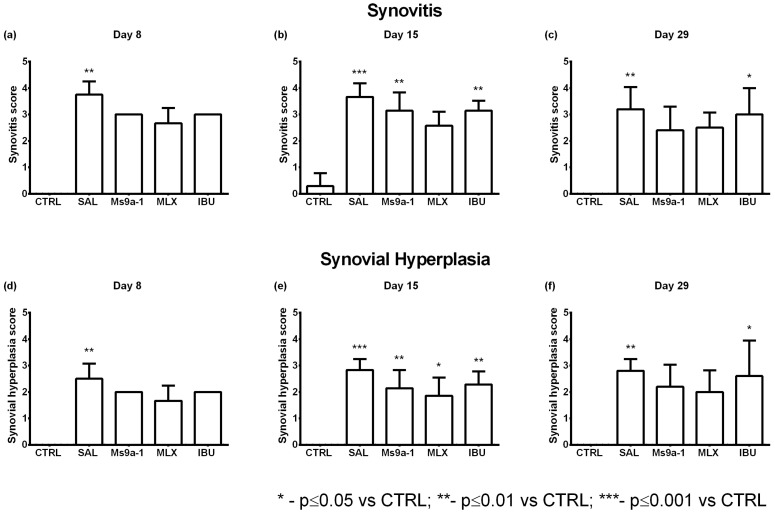
Histological analysis of synovitis and synovial hyperplasia of the arthritic knee joint in the MIA-induced osteoarthritis model. Synovitis and synovial hyperplasia were assessed on days 8 (**a**,**d**), 15 (**b**,**e**), and 29 (**c**,**f**) after intra-articular MIA injection. CTRL—control group, SAL—sterile saline, MLX—meloxicam, and IBU—ibuprofen. Results are shown as mean and SD (day 8, *n* = 3–4; day 15, *n* = 6–7; day 29, *n* = 4–5). Statistical analysis was performed using the Kruskal–Wallis test followed by Dunn’s multiple comparisons test. * *p* < 0.05 vs. CTRL, ** *p* < 0.01 vs. CTRL, *** *p* < 0.001 vs. CTRL.

**Figure 9 marinedrugs-21-00617-f009:**
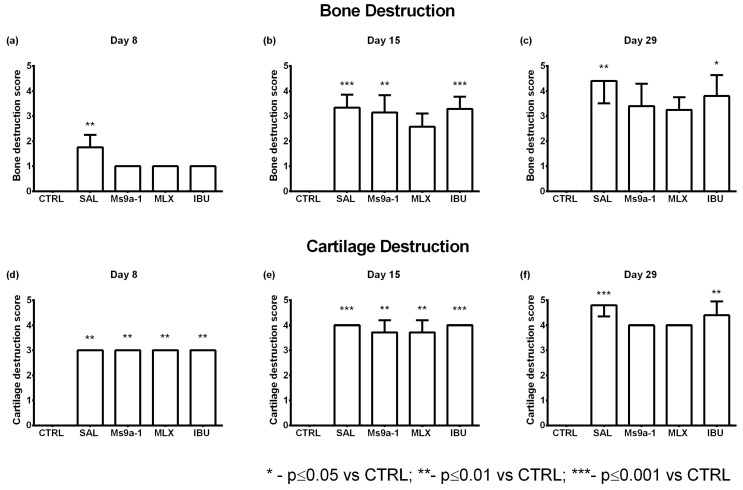
Histological analysis of cartilage and bone destruction of the arthritic knee joint in the MIA-induced osteoarthritis model. Bone and cartilage destruction were assessed on days 8 (**a**,**d**), 15 (**b**,**e**), and 29 (**c**,**f**) after intra-articular MIA injection. CTRL—control group, SAL—sterile saline, MLX—meloxicam, and IBU—ibuprofen. Results are shown as mean and SD (day 8, *n* = 3–4; day 15, *n* = 6–7; day 29, *n* = 4–5). Statistical analysis was performed using the Kruskal–Wallis test followed by Dunn’s multiple comparisons test. * *p* < 0.05 vs. CTRL, ** *p* < 0.01 vs. CTRL, *** *p* < 0.001 vs. CTRL.

**Figure 10 marinedrugs-21-00617-f010:**
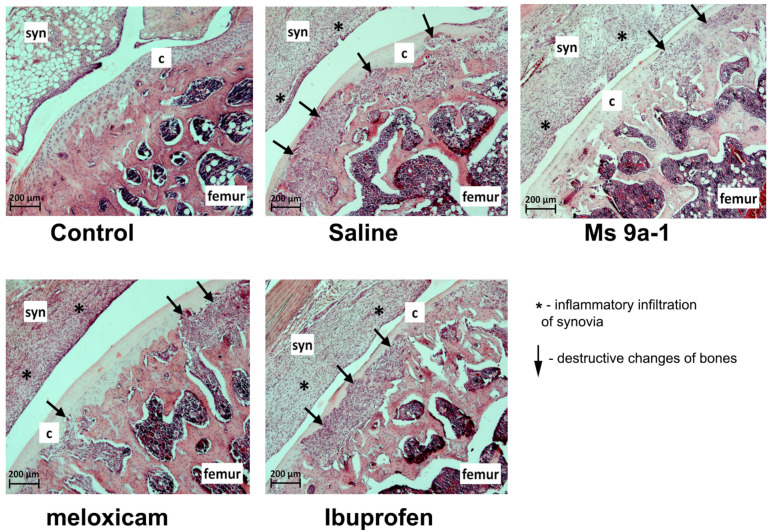
Representative images of histological analysis on day 15 after intra-articular MIA injection (scale bar = 200 µm, 50× magnification). Syn, synovia; C, cartilage; *, inflammatory infiltration of synovia; black arrows, destructive changes of bones. Each sign was graded with scores (0 to 5), where 0 represents normal tissue and 5 represents severe tissue degeneration. Control: Inflammatory infiltration (InIn) 0, synovial hyperplasia (SH) 0, cartlige destruction (CD) 0, bone destruction (BD) 0; sterile saline-treated group: InIn 4, SH 3,CD 4, BD 4; ibuprofen-treated group: InIn 3, SH 3, CD 4, BD 4; meloxicam-treated group: InIn 3, SH 2, CD 4, BD 3; Ms9a-1: InIn 3, SH 2, CD 4, BD 3.

## Data Availability

All data is contained within the article.

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
