# Peer review of "Potentiating TRPA1 by Sea Anemone Peptide Ms 9a-1 Reduces Pain and Inflammation in a Model of Osteoarthritis"

_marinedrugs, 2023, doi:10.3390/md21120617_

Round 1
Reviewer 1 Report
Comments and Suggestions for Authors
Manuscript "Potentiating TRPA1 by sea anemone peptide Ms 9a-1 reduces pain and inflammation in a model of osteoarthritis" shows important anti-inflammatory effects of the Ms 9a-1 peptide. However, the authors should clarify the following points:
- For the determination of anti-inflammatory activity, the most used pharmacological targets are interleukins, NF-KB, TNF-alpha, COX. The pharmacological target suggested by the authors was TRPA1, which is more related to analgesic effects. For this reason, I would like to know what the criteria were for choosing this drug target.
- What was the criterion for choosing the positive control APHC3 in the biological assays. I understand that since it is a peptide, it would be easier to suggest the possible mechanism that Ms 9a-1 would have, but I believe that there are other peptides that are directly related to pharmacological targets present in the inflammatory cascade.
- Because cytotoxicity tests of the Ms 9a-1 peptide were not carried out to determine what concentrations are safe.
Comments on the Quality of English Language
The manuscript has grammatical errors because the authors wrote the manuscript in parts. That is, there are fragments written in British English and other fragments are written in American English.
Author Response
We are grateful for your thorough review and criticism.
Manuscript "Potentiating TRPA1 by sea anemone peptide Ms 9a-1 reduces pain and inflammation in a model of osteoarthritis" shows important anti-inflammatory effects of the Ms 9a-1 peptide. However, the authors should clarify the following points:
- For the determination of anti-inflammatory activity, the most used pharmacological targets are interleukins, NF-KB, TNF-alpha, COX. The pharmacological target suggested by the authors was TRPA1, which is more related to analgesic effects. For this reason, I would like to know what the criteria were for choosing this drug target.
The role of TRPA1 in nerogenic inflammation and inflammatory diseases is well established (recent review, https://doi.org/10.3389/fphys.2023.1093925). We added to the introduction the text below to make this fact clear.
"The TRPA1 channel is involved in both the initiation/propagation of inflammatory responses and the detection of inflammatory mediators. Tissue damage leads to the release of inflammatory factors that cause activation or sensitization of the TRPA1 channel located at the endings of peptidergic nociceptive sensory neurons. The influx of Ca2+ causes the release of neuropeptides such as substance P (SP), neurokinin A (NKA), and calcitonin gene-related peptide (CGRP), initiating a number of signal-transduction pathways to promote the inflammatory process. Additionally, The depolarization of sensory neurons is perceived as pain in the CNS."
Moreover, Ms9a-1 showed a significant anti-inflammatory effect in CFA-induced inflammation, as described previously (https://doi.org/10.1074/jbc.M116.757369).
- What was the criterion for choosing the positive control APHC3 in the biological assays. I understand that since it is a peptide, it would be easier to suggest the possible mechanism that Ms 9a-1 would have, but I believe that there are other peptides that are directly related to pharmacological targets present in the inflammatory cascade.
Peptide APHC3 is additional control to NSAIDs. This peptide inhibits TRPV1 channel. It is strongly believed that TRPA1 always co-expresses with TRPV1 on sensory neurons. But not all TRPV1-positive sensory neurons express TRPA1. Therefore, it has some interest to compare desensitization of a portion of sensory neurons (TRPV1+/TRPA1+) by Ms9a-1 with inhibition of TRPV1 on all TRPV1+ nociceptors by АPHC3. These are molecules of similar class (peptides) acting on different channels involved in neurogenic inflammation.
We added to the discussion
"Peptide APHC3 inhibits the TRPV1 channel on sensory neurons and consequently reduces neurogenic inflammation. Thus, both APHC3 and Ms 9a-1 produce an effect on the excitability of sensory neurons via different molecular targets."
- Because cytotoxicity tests of the Ms 9a-1 peptide were not carried out to determine what concentrations are safe.
Ms9a-1 was found to be safe at a dose of 0.3 mg/kg. No changes in locomotor activity in the open-field test were found after i.v. administration of 0.3 mg/kg Ms 9a-1. All basic parameters of locomotion tests, such as traveled distance, rearing, velocity, and time spent in the center and peripheral zones were the same for the control and Ms 9a-1 groups (https://doi.org/10.1074/jbc.M116.757369).
We added this information to the introduction.
"Ms9a-1 was found to be safe; injection of peptide (10µL of ~70 µM solution) into the hindpaw did not cause pain, inflammation, or hypersensitivity, and no changes in locomotor activity in the open-field test were found after intravenous administration of 0.3 mg/kg."
The manuscript has grammatical errors because the authors wrote the manuscript in parts. That is, there are fragments written in British English and other fragments are written in American English.
We tried to correct the style. Two people wrote the most part of the manuscript that probably determined the difference in style.
Reviewer 2 Report
Comments and Suggestions for Authors
The manuscript “Potentiating TRPA1 by sea anemone peptide Ms 9a-1 reduces pain and inflammation in a model of osteoarthritis” by Maleeva et al, describes the effects of the peptide Ms 9a-1 from see anemone, which is a positive allosteric modulator of the TRPA1 receptor, in comparison with two anti-inflammatory drugs, usually used to treat osteoarthritis.
The manuscript is interesting and well conducted, even if some issues need to be addressed.
Major
The main point to be clarified is the number of animals used for each study group. The statistical analysis and in turn the significance of the whole manuscript depend on this parameter. Of course a limited number of rats could negatively/positively affect the results.
Minor
The authors should discuss the pharmacokinetic and the bioavailability of the peptide MS 9s-1 in Discussion section.
Comments on the Quality of English Language
Minor editing
Author Response
Thank you for the review!
The manuscript “Potentiating TRPA1 by sea anemone peptide Ms 9a-1 reduces pain and inflammation in a model of osteoarthritis” by Maleeva et al, describes the effects of the peptide Ms 9a-1 from see anemone, which is a positive allosteric modulator of the TRPA1 receptor, in comparison with two anti-inflammatory drugs, usually used to treat osteoarthritis.
The manuscript is interesting and well conducted, even if some issues need to be addressed.
Major
The main point to be clarified is the number of animals used for each study group. The statistical analysis and in turn the significance of the whole manuscript depend on this parameter. Of course a limited number of rats could negatively/positively affect the results.
The number of animals is shown in the figure legends. We specified the number more correctly in the revised version of the manuscript.
Minor
The authors should discuss the pharmacokinetic and the bioavailability of the peptide MS 9s-1 in Discussion section.
The pharmacokinetic and the bioavailability data on the peptide MS 9a-1 could be obtained in preclinical studies. Of course, we can suggest that sea anemone Cys-rich peptides have similar pharmakokinetic and bioavalability (and the most probably they do), but it is not correct to discuss it without the data.
Round 2
Reviewer 2 Report
Comments and Suggestions for Authors
The manuscript can be accepted in the present form, even if some comments on pharmacokinetic and bioavailability would have been appropriate